# Learning to Inpaint for Image Compression

**Mohammad Haris Baig**[*]
Department of Computer Science
Dartmouth College
Hanover, NH

**Vladlen Koltun**
Intel Labs
Santa Clara, CA

**Lorenzo Torresani**
Dartmouth College
Hanover, NH

## Abstract

We study the design of deep architectures for lossy image compression. We present two architectural recipes in the context of multi-stage progressive encoders and empirically demonstrate their importance on compression performance. Specifically, we show that: (a) predicting the original image data from residuals in a multi-stage progressive architecture facilitates learning and leads to improved performance at approximating the original content and (b) learning to inpaint (from neighboring image pixels) before performing compression reduces the amount of information that must be stored to achieve a high-quality approximation. Incorporating these design choices in a baseline progressive encoder yields an average reduction of over $60\%$ in file size with similar quality compared to the original residual encoder.

## 1 Introduction

Visual data constitutes most of the total information created and shared on the Web every day and it forms a bulk of the demand for storage and network bandwidth [13]. It is customary to compress image data as much as possible as long as there is no perceptible loss in content. In recent years deep learning has made it possible to design deep models for learning compact representations for image data [2, 16, 18, 19, 20]. Deep learning based approaches, such as the work of Rippel and Bourdev [16], significantly outperform traditional methods of lossy image compression. In this paper, we show how to improve the performance of deep models trained for lossy image compression.

We focus on the design of models that produce progressive codes. Progressive codes are a sequence of representations that can be transmitted to improve the quality of an existing estimate (from a previously sent code) by adding missing detail. This is in contrast to non-progressive codes whereby the entire data for a certain quality approximation must be transmitted before the image can be viewed. Progressive codes improve the user's browsing experience by reducing loading time of pages that are rich in images. Our main contributions in this paper are two-fold.

1. While traditional progressive encoders are optimized to compress residual errors in each stage of their architecture (residual-in, residual-out), instead we propose a model that is trained to predict at each stage the original image data from the residual of the previous stage (residual-in, image-out). We demonstrate that this leads to an easier optimization resulting in better image compression. The resulting architecture reduces the amount of information that must be stored for reproducing images at similar quality by $18\%$ compared to a traditional residual encoder.

2. Existing deep architectures do not exploit the high degree of spatial coherence exhibited by neighboring patches. We show how to design and train a model that can exploit dependences between adjacent regions by learning to inpaint from the available content. We introduce multi-scale convolutions that sample content at multiple scales to assist with inpainting.

---

[*]http://www.cs.dartmouth.edu/ haris/compression

We jointly train our proposed inpainting and compression models and show that inpainting reduces the amount of information that must be stored by an additional $42\%$.

## 2   Approach

We begin by reviewing the architecture and the learning objective of a progressive multi-stage encoder-decoder with $S$ stages. We adopt the convolutional-deconvolutional residual encoder proposed by Toderici et al. [19] as our reference model. The model extracts a compact binary representation $B$ from an image patch $P$. This binary representation, used to reconstruct an approximation of the original patch, consists of the sequence of representations extracted by the $S$ stages of the model, $B = [B_1, B_2, \ldots B_S]$.

The first stage of the model extracts a binary code $B_1$ from the input patch $P$. Each of the subsequent stages learns to extract representations $B_s$, to model the compressions residuals $R_{s-1}$ from the previous stage. The compression residuals $R_s$ are defined as $R_s = R_{s-1} - \mathcal{M}_s(R_{s-1}|\Theta_s)$, where $\mathcal{M}_s(R_{s-1}|\Theta_s)$ represents the reconstruction obtained by the stage $s$ when modelling the residuals $R_{s-1}$. The model at each stage is split into an encoder $B_s = \mathcal{E}_s(R_{s-1}|\Theta_s^E)$ and a decoder $\mathcal{D}_s(B_s|\Theta_s^D)$ such that $\mathcal{M}_s(R_{s-1}|\Theta_s) = \mathcal{D}_s(\mathcal{E}_s(R_{s-1}|\Theta_s^E)|\Theta_s^D)$ and $\Theta_s = \{\Theta_s^E, \Theta_s^D\}$. The parameters for the $s^{\text{th}}$ stage of the model are denoted by $\Theta_s$. The residual encoder-decoder is trained on a dataset $\mathcal{P}$, consisting of $N$ image patches, according to the following objective:

$$\hat{\mathcal{L}}(\mathcal{P}; \Theta_{1:S}) = \sum_{s=1}^{S} \sum_{i=1}^{N} \|R_{s-1}^{(i)} - \mathcal{M}_s(R_{s-1}^{(i)}|\Theta_s)\|_2^2. \tag{1}$$

$R_s^{(i)}$ represents the compression residual for the $i^{\text{th}}$ patch $P^{(i)}$ after stage $s$ and $R_0^{(i)} = P^{(i)}$.

Residual encoders are difficult to optimize as gradients have to traverse long paths from later stages to affect change in the previous stages. When moving along longer paths, gradients tend to decrease in magnitude as they get to earlier stages. We address this shortcoming of residual encoders by studying a class of architectures we refer to as "Residual-to-Image" (R2I).

### 2.1   Residual-to-Image (R2I)

To address the issue of vanishing gradients we add connections between subsequent stages and restate the loss to predict the original data at the end of each stage, thus performing *residual-to-image* prediction. This leads to the new objective shown below:

$$\mathcal{L}(\mathcal{P}; \Theta_{1:S}) = \sum_{s=1}^{S} \sum_{i=1}^{N} \|P^{(i)} - \mathcal{M}_s(R_{s-1}^{(i)}|\Theta_s)\|_2^2. \tag{2}$$

Stage $s$ of this model takes as input the compression residuals $R_{s-1}$ computed with respect to the original data, $R_{s-1} = P - \mathcal{M}_{s-1}(R_{s-2}|\Theta_{s-1})$, and $\mathcal{M}_{s-1}(R_{s-2}|\Theta_{s-1})$ now approximates the reconstruction of the original data $P$ at stage $s-1$. To allow complete image reconstructions to be produced at each stage while only feeding in residuals, we introduce connections between the layers of adjacent stages. These connections allow for later stages to incorporate information that has been recovered by earlier stages into their estimate of the original image data. Consequently, these connections (between subsequent stages) allow for better optimization of the model.

In addition to assisting with modeling the original image, these connections play two key roles. Firstly, these connections create residual blocks [10] which encourage explicit learning of how to reproduce information which could not be generated by the previous stage. Secondly, these connections reduce the length of the path along which information has to travel from later stages to impact the earlier stages, leading to a better joint optimization.

This leads us to the question of where should such connections be introduced and how should information be propagated? We consider two types of connections to propagate information between successive stages. 1) *Prediction connections* are analogous to the identity shortcuts introduced by He et al. [10] for residual learning. They act as parameter-free additive connections: the output of

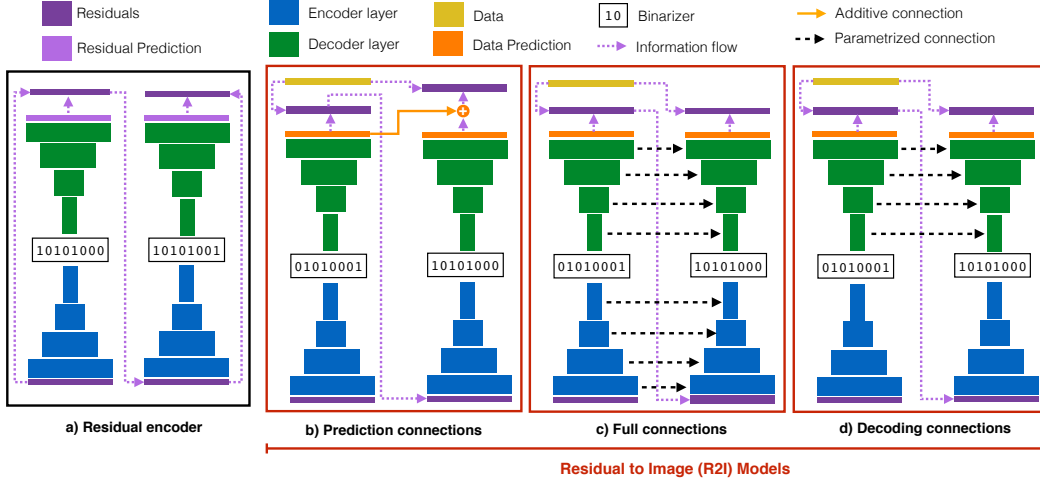

Figure 1: Multiple approaches for introducing connections between successive stages. These designs for progressive architectures allow for varying degrees of information to be shared. Architecture (b-d) do not reconstruct residuals, but the original data at every stage. We call these architectures "residual-to-image" (R2I).

each stage is produced by simply adding together the residual predictions of the current stage and all preceding stages (see Figure 1(b)) before applying a final non-linearity.2) *Parametric connections* are referred to as projection shortcuts by He et al. [10]. Here we use them to connect corresponding layers in two consecutive stages of the compression model. The features of each layer from the previous stage are convolved with learned filters before being added to the features of the same layer in the current stage. A non-linearity is then applied on top. The prediction connections only yield the benefit of creating residual blocks, albeit very large and thus difficult to optimize. In contrast, parametric connections allow for the intermediate representations from previous stages to be passed to the subsequent stages. They also create a denser connectivity pattern with gradients now moving along corresponding layers in adjacent stages. We consider two variants of parametric connections: "full" which use parametric connections between all the layers in two successive stages (see Figure 1(c)), and "decoding" connections which link only corresponding decoding layers (i.e., there are no connections between encoding layers of adjacent stages). We note that the LSTM-based model of Toderici et al. [20] represents a particular instance of R2I network with full connections. In Section 3 we demonstrate that R2I models with decoding connections outperform those with full connections and provide an intuitive explanation for this result.

## 2.2 Inpainting Network

Image compression architectures learn to encode and decode an image patch-by-patch. Encoding all patches independently assumes that the regions contain truly independent content. This assumption generally does not hold true when the patches being encoded are contiguous. We observe that the content of adjacent image patches is not independent. We propose a new module for the compression model designed to exploit the spatial coherence between neighboring patches. We achieve this goal by training a model with the objective of predicting the content of each patch from information available in the neighboring regions.

Deep models for inpainting, such as the one proposed by Pathak et al. [14], are trained to predict the values of pixels in the region $\hat{W}$ from a context region $\hat{C}$ (as shown in Figure 2). As there is data present all around the region to be inpainted this imposes strong constraints on what the inpainted region should look like. We consider the scenario where images are encoded and decoded block-by-block moving from left to right and going from top to bottom (similar to how traditional codecs process images [1, 21]). Now, at decoding time only content above and to the left of each patch will have been reconstructed (see Figure 2(a)). This gives rise to the problem of "partial-context inpainting". We propose a model that, given an input region $C$, attempts to predict the content of the current patch $P$. We denote by $\hat{\mathcal{P}}$ the dataset which contains all the patches from the dataset $\mathcal{P}$

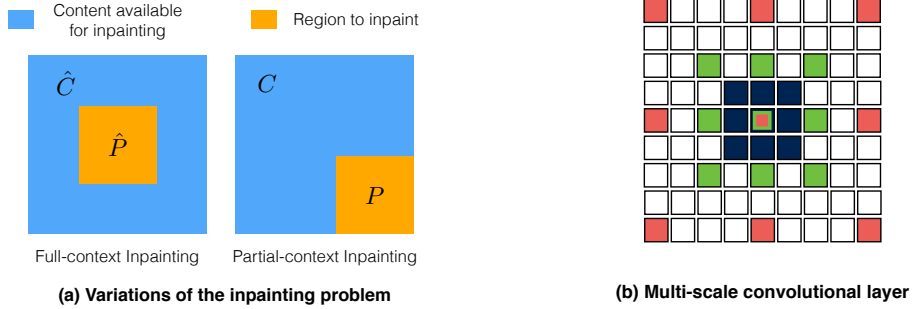

**(a) Variations of the inpainting problem**        **(b) Multi-scale convolutional layer**

Figure 2: (a) The two kinds of inpainting problems. (b) A multi-scale convolutional layer with 3 dilation factors. The colored boxes represent pixels from which the content is sampled.

and the respective context regions $C$ for each patch. The loss function used to train our inpainting network is:

$$\mathcal{L}_{inp}(\hat{\mathcal{P}}; \Theta_I) = \sum_{i=1}^{N} \|P^{(i)} - \mathcal{M}_I(C^{(i)}|\Theta_I)\|_2^2. \tag{3}$$

The output of the inpainting network is denoted by $\mathcal{M}_I(C^{(i)}|\Theta_I)$, where $\Theta_I$ refers to the parameters of the inpainting network.

### 2.2.1 Architecture of the Partial-Context Inpainting Network

Our inpainting network has a feed-forward architecture which propagates information from the context region $C$ to the region being inpainted, $P$. To improve the ability of our model at predicting content, we use a multi-scale convolutional layer as the basic building block of our inpainting network. We make use of the dilated convolutions described by Yu and Koltun [23] to allow for sampling at various scales. Each multi-scale convolutional layer is composed of $k$ filters for each dilation factor being considered. Varying the dilation factor of filters gives us the ability to analyze content at various scales. This structure of filters provides two benefits. First, it allows for a substantially denser and more diverse sampling of data from context and second it allows for better propagation of content at different spatial scales. A similarly designed layer was also used by Chen et al. [5] for sampling content at multiple scales for semantic segmentation. Figure 2(b) shows the structure of a multi-scale convolutional layer.

The multi-scale convolutional layer also gives us the freedom to propagate content at full resolution (no striding or pooling) as only a few multi-scale layers suffice to cover the entire region. This allows us to train a relatively shallow yet highly expressive architecture which can propagate fine-grained information that might otherwise be lost due to sub-sampling. This light-weight and efficient design is needed to allow for joint training with a multi-stage compression model.

### 2.2.2 Connecting the Inpainting Network with the R2I Compression model

Next, we describe how to use the prediction of the inpainting network for assisting with compression. Whereas the inpainting network learns to predict the data as accurately as possible, we note that this is not sufficient to achieve good performance on compression, where it is also necessary that the "inpainting residuals" be easy to compress. We describe the inpainting residuals as $R_0 = P - \mathcal{M}_I(C|\Theta_I)$, where $\mathcal{M}_I(C|\Theta_I)$ denotes the inpainting estimate. As we wanted to train our model to always predict the data, we add the inpainting estimate to the final prediction of each stage of our compression model. This allows us to (a) produce the original content at each stage and (b) to

discover an inpainting that is beneficial for all stages of the model because of joint training. We now train our complete model as

$$\mathcal{L}_C(\hat{\mathcal{P}}; \Theta_I, \Theta_{1:S}) = \mathcal{L}_{inp}(\hat{\mathcal{P}}; \Theta_I) + \sum_{i=1}^{N} \sum_{s=1}^{S} \| P^{(i)} - [\mathcal{M}_s(R_{s-1}^{(i)} | \Theta_s) + \mathcal{M}_I(C^{(i)} | \Theta_I)] \|_2^2. \quad (4)$$

In this new objective $\mathcal{L}_C$, the first term $\mathcal{L}_{inp}$ corresponds to the original inpainting loss, $R_0^{(i)}$ corresponds to the inpainting residual for example $i$. We note that each stage of this inpainting-based progressive coder directly affects what is learned by the inpainting network. We refer to the model trained with this joint objective as "Inpainting for Residual-to-Image Compression" (IR2I).

Whereas we train our model to perform inpainting from the original image content, we use a lossy approximation of the context region $C$ when encoding images with IR2I. This is done because at decoding time our model does not have access to the original image data. We use the approximation from stage 2 of our model for performing inpainting at encoding and decoding time, and transmit the binary codes for the first two stages as a larger first code. This strategy allows us to leverage inpainting while performing progressive image compression.

## 2.3 Implementation Details

Our models were trained on $6,507$ images from the ImageNet dataset [7], as proposed by Ballé et al. [2] to train their single-stage encoder-decoder architectures. A full description of the R2I models and the inpainting network is provided in the supplementary material. We use the Caffe library [11] to train our models. The residual encoder and R2I models were trained for $60,000$ iterations whereas the joint inpainting network was trained for $110,000$ iterations. We used the Adam optimizer [12] for training our models and the MSRA initialization [9] for initializing all stages. We used initial learning rates of $0.001$ and the learning rate was dropped after 30K and 45K for the R2I models. For the IR2I model, the learning rate was dropped after 30K, 65K, and 90K iterations by a factor of 10 each time. All of our models were trained to reproduce the content of $32 \times 32$ image patches. Each of our models has 8 stages, with each stage contributing $0.125$ bits-per-pixel (bpp) to the total representation of a patch. Our models handle binary optimization by employing the biased estimators approach proposed by Raiko et al. [15] as was done by Toderici et al. [19, 20].

Our inpainting network has 8 multi-scale convolutional layers for content propagation and one standard convolutional layer for performing the final prediction. Each multi-scale convolutional layer consists of 24 filters each for dilation factors $1, 2, 4, 8$. Our inpainting network takes as input a context region $C$ of size $64 \times 64$, where the bottom right $32 \times 32$ region is zeroed out and represents the region to be inpainted.

## 3 Results

We investigate the improvement brought about by the presented techniques. We are interested in studying the reduction in bit-rate, for varying quality of reconstruction, achieved after adaptation from the residual encoder proposed by Toderici et al. [19]. To evaluate performance, we perform compression with our models on images from the Kodak dataset [8]. The dataset consists of 24 uncompressed color images of size $512 \times 768$. The quality is measured according to the MS-SSIM [22] metric (higher values indicate better quality). We use the Bjontegaard-Delta metric [4] to compute the average reduction in bit-rate across all quality settings.

### 3.1 R2I - Design and Performance

The table in Figure 3(a) shows the percentage reduction in bit-rate achieved by the three variations of the Residual-to-Image models. As can be seen, adding side-connections and training for the more desirable objective (i.e., approximating the original data) at each stage helps each of our models. That said, having connections in the decoder only helps more compared to using a "full" connection approach or only sharing the final prediction.

| | Rate Savings (%) | |
|---|---|---|
| Approach | SSIM | MS-SSIM |
| R2I Prediction | 4.483 | 5.177 |
| R2I Full | 10.015 | 7.652 |
| R2I Decoding | **20.002** | **17.951** |

(a)

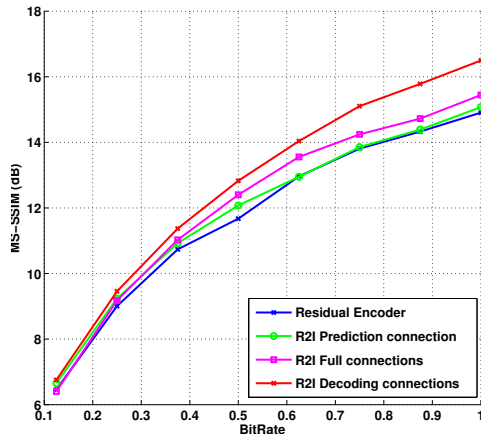

(b)

Figure 3: (a) Average rate savings for each of the three R2I variants compared to the residual encoder proposed by Toderici et al. [19]. (b) Figure shows the quality of images produced by each of the three R2I variants across a range of bit-rates.

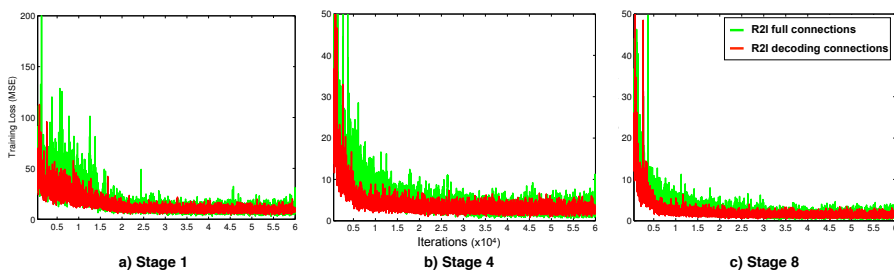

Figure 4: The R2I training loss from 3 different stages (start, middle, end) viewed as a function of iterations for the "full" and the "decoding" connections models. We note that the decoding connections model converges faster, to a lower value, and shows less variance.

The model which shares only the prediction between stages performs poorly in comparison to the other two designs as it does not allow for features from earlier stages to be altered as efficiently as done by the full or decoding connections architectures.

The model with decoding connections does better than the architecture with full connections because for the model with connections at decoding only the binarization layer in each stage extracts a representation from the relevant information only (the residuals with respect to the data). In contrast, when connections are established in both the encoder and the decoder, the binary representation may include information that has been captured by a previous stage, thereby adding burden on each stage in identifying information pertinent to improving reconstruction, leading to a tougher optimization. Figure 4 shows that the model with full connections struggles to minimize the training error compared to the model with decoding connections. This difference in training error points to the fact that connections in the encoder make it harder for the model to do well at training time. This difficulty of optimization amplifies with the increase in stages as can be seen by the difference between the full and decoding architecture performance (shown in Figure 3(b)) because the residuals become harder to compress.

We note that R2I models significantly improve the quality of reconstruction at higher bit rates but do not improve the estimates at lower bit-rates as much (see Figure 3(b)). This tells us that the overall performance can be improved by focusing on approaches that yield a significant improvement at lower bit-rates, such as inpainting, which is analyzed next.

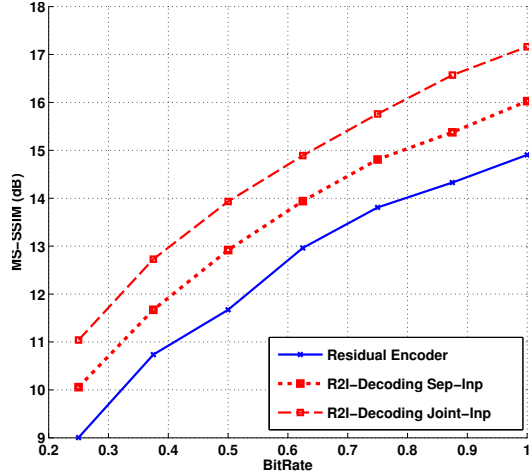

| Approach | Rate Savings (%) | |
| --- | --- | --- |
| | SSIM | MS-SSIM |
| R2I Decoding | 20.002 | 17.951 |
| R2I Decoding Sep-Inp | 27.379 | 27.794 |
| R2I Decoding Joint-Inp | **63.353** | **60.446** |

(a) Impact of inpainting on the performance at compression. All bit-rate savings are reported with respect to the residual encoder by Toderici et al. [19]

(b)

Figure 5: (a) Average rate savings with varying forms of inpainting. (b) The quality of images with each of our proposed approaches at varying bit-rates.

## 3.2    Impact of Inpainting

We begin analyzing the performance of the inpainting network and other approaches on partial-context inpainting. We compare the performance of the inpainting network with both traditional approaches as well as a learning-based baseline. Table 1 shows the average SSIM achieved by each approach for inpainting all non-overlapping patches in the Kodak dataset.

| Approach | PDE-based | Exemplar-based | Learning-based | |
| --- | --- | --- | --- | --- |
| | [3] | [6] | Vanilla network | Inpainting network |
| SSIM | 0.4574 | 0.4611 | 0.4545 | **0.5165** |

Table 1: Average SSIM for partial-context inpainting on the Kodak dataset [8]. The vanilla model is a feed-forward CNN with no multi-scale convolutions.

The vanilla network corresponds to a 32-layer (4 times as deep as the inpainting network) model that does not use multi-scale convolutions (all filters have a dilation factor of 1), has the same number of parameters, and also operates at full resolution (as our inpainting network). This points to the fact that the improvement in performance of the inpainting network over the vanilla model is a consequence of using multi-scale convolutions. The inpainting network improves over traditional approaches because our model learns the best strategy for propagating content as opposed to using hand-engineered principles of content propagation. The low performance of the vanilla network shows that learning by itself is not superior to traditional approaches and multi-scale convolutions play a key role in achieving better performance.

Whereas inpainting provides an initial estimate of the content within the region it by no means generates a perfect reconstruction. This leads us to the question of whether this initial estimate is better than not having an estimate? The table in Figure 5(a) shows the performance on the compression task with and without inpainting. These results show that the greatest reduction in file size is achieved when the inpainting network is jointly trained with the R2I model. We note (from Figure 5(b)) that inpainting greatly improves the quality of results obtained at lower and at higher bit rates.

The baseline where the inpainting network is trained separately from the compression network is presented here to emphasize the role of joint training. Traditional codecs [1] use simple non learning-based inpainting approaches and their predefined methods of representing data are unable to compactly encode the inpainting residuals. Learning to inpaint separately improves the performance

as the inpainted estimate is better than not having any estimate. But given that the compression model has not been trained to optimize the compression residuals the reduction in bit-rate for achieving high quality levels is low. We show that with joint training, we can not only train a model that does better inpainting but also ensure that the inpainting residuals can be represented compactly.

### 3.3 Comparison with Existing Approaches

Table 2 shows a comparison of the performance of various approaches compared to JPEG [21] in the 0.125 to 1 bits-per-pixel (bpp) range. We select this range as images from our models towards the end of this range show no perceptible artifacts of compression.

The first part of the table evaluates the performance of learning-based progressive approaches. We note that our proposed model outperforms the multi-stage residual encoder proposed by Toderici et al. [19] (trained on the same 6.5K dataset) by 17.9% and IR2I outperforms the residual encoder by reducing file-sizes by 60.4%. The residual-GRU, while similar in architecture to our "full" connections model, does not do better even when trained on a dataset that is 1000 times bigger and for 10 times more training time. The results shown here do not make use of entropy coding as the goal of this work is to study how to improve the performance of deep networks for progressive image compression and entropy coding makes it harder to understand where the performance improvements are coming from. As various approaches use different entropy coding methods, this further obfuscates the source of the improvements.

The second part of the table shows the performance of existing codecs. Existing codecs use entropy coding and rate-distortion optimization. We note that even without using either of these powerful post processing techniques, our final "IR2I" model is competitive with traditional methods for compression, which use both of these techniques. A comparison with recent non-progressive approaches [2, 18], which also use these post-processing techniques for image compression, is provided in the supplementary material.

| Approach | Number of Training Images | Progressive | Rate Savings (%) |
|---|---|---|---|
| Residual Encoder [19] | 6.5K | Yes | 2.56 |
| Residual-GRU [20] | 6M | Yes | 33.26 |
| R2I (Decoding connections) | 6.5K | Yes | 18.53 |
| IR2I | 6.5K | Yes | **51.25** |
| JPEG-2000 [17] | N/A | No | 63.01 |
| WebP [1] | N/A | No | 64.98 |

Table 2: Average rate savings compared to JPEG [21]. The savings are computed on the Kodak [8] dataset with rate-distortion profiles measuring MS-SSIM in the 0-1 bpp range.

We observe that a naive implementation of IR2I creates a linear dependence in content (as all regions used as context have to be decoded before being used for inpainting) and thus may be substantially slower. In practice, this slowdown would be negligible as one can use a diagonal scan pattern (similar to traditional codecs) for ensuring high parallelism thereby reducing run times. Furthermore, we perform inpainting using predictions from the first step only. Therefore, the dependence only exists when generating the first progressive code. For all subsequent stages, there is no dependence in content, and our approach is comparable in run time to similar approaches.

## 4 Conclusion and Future Work

We study a class of "Residual to Image" models and show that within this class, architectures which have decoding connections perform better at approximating image data compared to designs with other forms of connectivity. We observe that our R2I decoding connections model struggles at low bit-rates and we show how to exploit spatial coherence between content of adjacent patches via inpainting to improve performance at approximating image content at low bit-rates. We design a new

model for partial-context inpainting using multi-scale convolutions and show that the best way to leverage inpainting is by jointly training the inpainting network with our R2I Decoding model.

One interesting extension of this work would be to incorporate entropy coding within our progressive compression framework to train models that produce binary codes which have low-entropy and can be represented even more compactly. Another possible direction would be to extend our proposed framework to video data, where the gains from our discovery of recipes for improving compression may be even greater.

# 5 Acknowledgements

This work was funded in part by Intel Labs and NSF award CNS-120552. We gratefully acknowledge NVIDIA and Facebook for the donation of GPUs used for portions of this work. We would like to thank George Toderici, Nick Johnston, Johannes Balle for providing us with information needed for accurate assessment. We are grateful to members of the Visual Computing Lab at Intel Labs, and members of the Visual Learning Group at Dartmouth College for their feedback.

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
