[Reviews · NeurIPS 2017]

Reviewer 1



This paper proposes a progressive image compression method that's "hybrid". The authors use the framework of Toderici et al (2016) to setup a basic progressive encoder, and then they improve on it by studying how to better propagate information between iterations. The solution involves using "temporal" residual connections, without the explicit need to have an RNN per se (though this point is a bit debatable because in theory if the residual connections are transformed by some convolution, are they acting as an additive RNN or not?). However, the authors also employ a predictor (inpainter). this allows them to encode each patch after trying to predict an "inpainted" version first. It is important to note here that this will only work (in practice) if all the patches on which this patch depends on have been decoded. This introduces a linear dependency on patches, which may make the method too slow in practice, and it would be nice to see a bit more in the text about this issue (maybe some timing in formation vs. not using inpainting). Overall, I think the paper was well written and an expert should be able to reproduce the work. Given that the field of neural image compression is still in its infancy and that most of the recent papers have been focusing on non-progressive methods, and this paper proposes a *progressive* encoder/decoder, I think we should seriously consider accepting it.

Reviewer 2



Paper modifies a version of auto encoder that can progressively compress images, by reconstructing the full image rather then residual at every state. This improves performance in low loss regime. In addition they learn an in paining network that learns to predict an image patches sequentially from previous patches that are located to the left and above of them. This results in the improved performance overall. Advantages: Interesting experiments and observations and improvement over previous residual encoder. Drawbacks: Why is the network applied in patches rather then over the whole image since it is convolutional network? The inpainting predictions are unimodal - whereas the distribution of the next patch is highly multimodal - this produces limitations and can be significantly improved. The resulting performance therefore is not state of the art (compared to standard non-progressive method like jpeg 2000). Other comments: - Have you tried feeding the image into encoder at every stage?

Reviewer 3



The authors take a model from the paper Toderici et al. [18] for the task of image compression and add their ideas to improve the compression rate. The proposal that the authors make to this model are: 1. To provide an identity shorcut from the previous stage to the output of the current stage 2. Add projection shortcuts to Encoder and/or Decoder from their previous stage 3. Inpainting model (with multi-scale convolution layers) at the end of each stage. The authors provide many different experiments (with/without inpainting network) and observe that their model with projection shortcuts only for the Decoder and jointly trained with an inpainting network provides for the best performance. Particularly, the inpainting network(by using multi-scale convolutions), as the authors claim, helps in improving the performance significantly at lower bit-rates. The paper shows many results to claim that the Decoder with parametric connections perform better than their other models. The reviewer finds it unclear with their argument that adding connections to Encoders would burden it on each stage. Perhaps an additional experiment / more explanation might give insights on how and why the Encoder with connections make the residuals harder to compress. Typos: Line 231 - "better then" -> "better than" It is an interesting paper with several experiments to validate their ideas.